# AXIOMATIZATION OF CONCEPT CNN EXPLANATIONS

## ABSTRACT

Concept-based explanations for convolutional neural networks (CNNs) offer human-interpretable insights into the decision-making processes of artificial intelligence (AI) models. In contrast to attribution-based methods, which primarily highlight salient pixels, concept-based approaches capture higher-level semantic features, thereby elucidating not only where the model looked but also what it saw. Despite their promise, the absence of rigorous axiomatic foundations has impeded systematic evaluation, comparison, and compliance, limiting their broader adoption. This paper presents a conceptual axiomatic framework, derived from the principles of explanation logic, for evaluating the faithfulness of concept-based explanations in CNN-driven image classification. We propose a novel set of axioms that formalize essential criteria for trustworthy explanations and establish a quantitative methodology for their evaluation. Extensive experiments conducted in both ideal and adversarial settings, across diverse model architectures, demonstrate the necessity and validity of these axioms. Our findings contribute to the development of reliable, interpretable, and trustworthy explainable artificial intelligence (XAI) frameworks, with particular relevance to high-stakes domains where transparent decision-making is crucial.

## 1 INTRODUCTION

Convolutional neural networks (CNN) have achieved remarkable success in computer vision Upadhyay et al. (2025). However, their black-box nature raises concerns about transparency Lincoln IV (2025). This requires understanding the CNN logic through post hoc explanation methods Akpudo et al. (2025b); Upadhyay et al. (2025).

Concept-based explanation for a CNN seeks to address the question: *Why did a CNN assign an input to a class?* This aligns with findings from cognitive science Akpudo et al. (2025b); Upadhyay et al. (2025); Moore et al. (2024), which suggest that among the many potential influences, humans generally expect explanations to highlight the key concepts behind an outcome. These concepts (predefined or discovered automatically) represent *high-level patterns or abstract ideas within an image class* that contribute to a class prediction Kim et al. (2018) (see Figure 1). However, currently there is no consensus on how to evaluate XAI Nauta et al. (2023b); Akpudo et al. (2025b).

The prevalent approaches, including concept-based methods **?**Zhang et al. (2021); Fel et al. (2023); Ghorbani et al. (2019), often rely on "cherry-picked" examples that appear intuitive, which many argue are anecdotal, inadequate, and potentially misleading He et al. (2025). However, unverified intuition can facilitate misapprehension Akpudo et al. (2025b); Lincoln IV (2025). While qualitative axioms rely on expert intervention to address inherent biases and are not covered within the scope of our study, quantitative axioms ensure objective transparency Kim et al. (2023). Unfortunately, the lack of sufficient quantitative evaluation hinders progress in interpretability research, as anecdotal inspection fails to verify concept explanations, thus undermining trust Upadhyay et al. (2025); He et al. (2025).

*Concept explanations* should address logical questions and satisfy desirable properties, that is, the axiomatic foundations of their design and evaluation Amgoud & Ben-Naim (2022); Sundararajan et al. (2017); Chen et al. (2023). For example, *How simple are the explanations? Are they coherent? How sane is the explainer?* etc. However, state-of-the-art concept-based explainers do not explicitly address these questions, prompting an essential inquiry: *What axiomatic foundations guarantee faithful and trustworthy concept explanations?* As illustrated in Figure 1, establishing

Figure 1: **Establishing Axiomatic Foundations for Concept-Based Explanations**. The explanations highlight different concepts (*Ear*, *Eye*, *Nose*, *Cheek*, and ***Background***) and their accompanying *prototypes* for a dog class. However, there is a need to answer questions like: how *simple* are the explanations? Are they *coherent*? how *sane* is the explainer? is the ***Background*** concept *relevant* or *causal* to the dog prediction?, etc. A sufficient set of axiomatic foundations provides answers and ensures comprehensive and faithful evaluation of *concept explanations*, fostering trust in the explainer.

robust axiomatic foundations requires *coreness*[1] and demands considering factors such as cognition, perception, and the ethical implications related to the design, development and adoption of CNNs in AI-based systems Akpudo et al. (2025b); Lincoln IV (2025); Upadhyay et al. (2025).

The paper's contributions are as follows: (**1**) Our work introduces key axioms that form a unified axiomatic foundation for concept-based explanations of CNNs. The axioms establish a cohesive set of quantitative checkpoints to evaluate explainer performance. (**2**) This study highlights the need for axioms that reduce uncertainty about label confirmation, serving as diversified and unified benchmarks to foster user confidence and trust. Our cross-model investigations emphasize blind, fair, and transparent evaluations within unified frameworks. (**3**) Comprehensive quantitative analysis confirms the importance of axiomatic compliance of concept-based explainers under ideal conditions and adversarial (Frontdoor[2] and Poisoning[3]) attacks, underscoring their role in building and maintaining trust. (**4**) delivers critical insights that enhance transparency in AI systems while maintaining high levels of performance and resilience in CNN-based classification, where transparency is crucial.

## 2 PROPOSED METHOD

Figure 2 illustrates the rationale for the concept-based explanation of a CNN $f(\cdot)$ composed of convolutional layers with rectified linear unit activations (Conv-ReLU) $E(\cdot)$ and a classifier $C(\cdot)$. For an *instance*[4] $x \in \mathcal{X}$ with label $y \in \mathcal{Y}$, the CNN executes a classification task $f(\cdot) : \mathcal{X} \rightarrow \mathcal{Y}$ such that $f(\cdot) = C(E(\cdot))$. The encoder $E(\cdot)$ produces a feature map $\mathcal{A} \in \mathbb{R}^{m \times c}$ in the penultimate layer, with $m = (h, w)$ denoting spatial di-

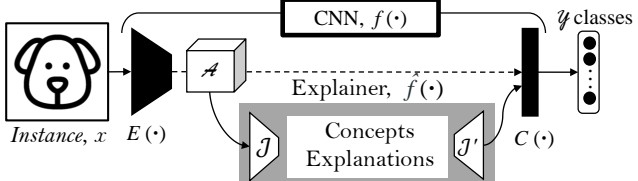

Figure 2: Concept-based explanation rationale for a CNN. The explainer $\hat{f}\{\mathcal{J}, \mathcal{J}'\}$ exploits $\mathcal{A}$ at the penultimate layer to generate *concept explanations*.

mensions and $c$ the number of channels containing discriminative information. The classifier $C(\cdot)$ maps $\mathcal{A}$ to the label space $\mathcal{Y}$ with trainable weights $t$. A concept-based explainer $\hat{f}\{\mathcal{J}, \mathcal{J}'\}$ constitutes the encoder $\mathcal{J}$ and the decoder $\mathcal{J}'$.

Figure 3 shows the proposed axiomatic concept-based explanation framework for CNNs. Given $x$ and $f(\cdot)$, $\mathcal{J}$ encodes $\mathcal{A}$ to generate concepts $\mathcal{S} \in \mathbb{R}^{m \times c'}$ and a fixed concept activation vectors

---

[1]The extent to which an explanation captures the most informative, non-redundant *concept explanations*.
[2]Early-stage image perturbations in the data pipeline.
[3]Training-time parameter attacks via noise, backdoors, or biased modifications.
[4]An *instance* is an image used to assess a concept's influence on a CNN's inference.

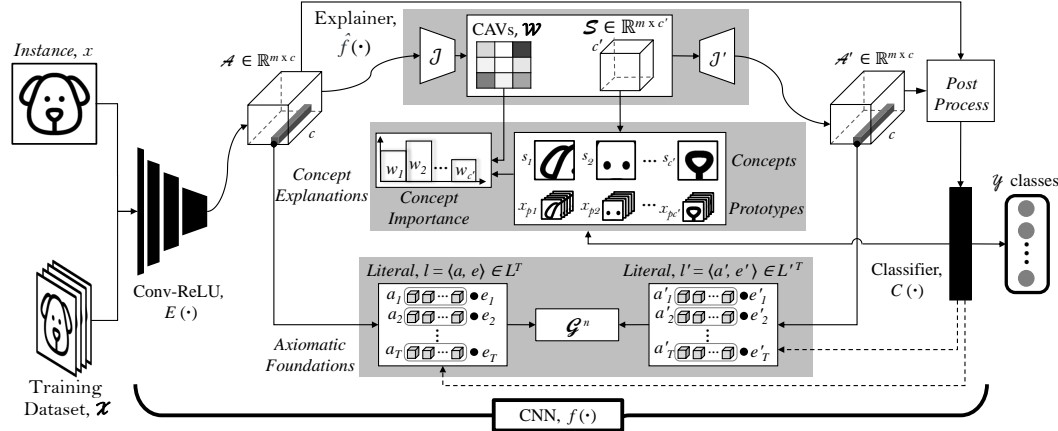

Figure 3: Schematic for the proposed axiomatic concept-based explanation framework for CNNs. The Conv-ReLU $E(\cdot)$ generates the featuremap $\mathcal{A}$ from the image *instance* at the CNN penultimate layer. The encoder $\mathcal{J}$ generates the concepts $\mathcal{S} \in \mathbb{R}^{m \times c'}$ and CAVs $\mathcal{P} \in \mathbb{R}^{c \times c'}$ from the featuremap $\mathcal{A} \in \mathbb{R}^{m \times c}$ (containing *literals* $l \in L$) produced by the CNN's Conv-ReLU $E(\cdot)$. The decoder $\mathcal{J}'$ produces $\mathcal{A}' \approx \mathcal{A}$. The classifier $C(\cdot)$ quantifies *concept importance* $\mathcal{W}$ for *concept explanation* for evaluating faithfulness using the axiomatic foundations $\mathcal{G}^n$. Post-processing methods such as *Flatten, Global Average Pooling, or Attention mechanisms* depend on the explainer's objective.

(CAV) $\mathcal{P} \in \mathbb{R}^{c \times c'}$ such that $\mathcal{J}(\mathcal{A}) = \mathcal{S}\mathcal{P} + \epsilon$, where $c'$ is a user-defined number of concepts, $\epsilon$ is the decomposition error and $c' \ll c$. Conversely, $\mathcal{J}'(\mathcal{S})$ decodes $\mathcal{S} \to \mathcal{A}'$(where $\mathcal{A}' \approx \mathcal{A}$) using the fixed $\mathcal{P}$. A *concept explanation* aims to generate discriminative information, i.e., concepts stored in $\mathcal{S}$, accompanied by their corresponding importance scores $\mathcal{W}$ for the class $y$ Zhang et al. (2021).

Our proposal is based on the non-triviality and minimal sufficiency of *concept explanations* and requires formal axiomatic principles (see Section 2.2). We hypothesize that $\{C(\mathcal{A}), C(\mathcal{A}')\}_{i=1}^n$ should constitute the axiomatic foundations $\mathcal{G}^n$, providing a principled basis to assess the explainer.

## 2.1 PROBLEM STATEMENT

We probe with a *class question*[5] $\mathbf{Q} = \langle \mathcal{S}, R, x \rangle$, the decision logic of $f(\cdot)$, seeking to evaluate the *reasons*[6] for $f(x)$, $\mathbf{R} = \langle \mathcal{S}, \mathbf{x}', \mathcal{W}, \mathcal{Y} \rangle$ that defines the mapping $\hat{f}_{\mathbf{R}} : \mathcal{S} \to \mathcal{Y}$ if $\hat{f}_{\mathbf{R}}(s) = y$ for $s \in \mathcal{S}$ using the proposed set of axiomatic foundations. These *reasons* constitute three key ingredients:

1. *literals* $l \in L$, formed by pairing feature activations $a \in \mathcal{A}$ with their concept activations $e \in C(a)$. Given a triple $T = \langle \mathcal{A}, f, \mathcal{Y} \rangle$, a *literal* $l \in L$ on $T$ is a couple $l = \langle a, e \rangle$ such that $a \in \mathcal{A}$, $e \in C(a)$. $L^T$ is the set of all literals, and $L$ is a subset of $L^T$.

2. *concept importance* $\mathcal{W} = \mathcal{P} \cdot t$ estimated as the sensitivity of $f(x)$ to $\mathcal{S}$ along $\mathcal{P}$ Kim et al. (2018).

3. *prototypes* $x_p$, an *instance* $x' \in \mathcal{X}_y$ whose *literals* $L'$ align maximally with the *literals* $L$ of a given *instance* $x$ of a class $y \in \mathcal{Y}$ such that $x_p = \arg\max_{x \in \mathcal{X}_y} \mathbb{E}_{x' \sim P(\mathcal{X}_y)} \phi(L, L')$, where $\mathbb{E}_{x' \sim P(\mathcal{X}_y)} \phi(L, L')$ denotes the expectation over $x'$ drawn from the probability distribution $P(\mathcal{X}_y)$ and $\phi(L, L')$ measures the homogeneity. Reliable homogeneity measures include Jaccard, Cosine, and Kernel-based similarity metrics Xie et al. (2016); Chen et al. (2019); Fel et al. (2023); Ma et al. (2023) (See **Prototype Selection** in Appendix).

---

[5]For an *instance* $x \in \mathcal{X}_y$, a *class question* $\mathbf{Q}$ is an abstraction of *infinite* possible queries that aims to identify the *reasons* that influenced the decision $f(x) = y$. $\hat{f}(\mathbf{Q})$ represents the set of possible explanations for $\mathbf{Q}$ which include contrastive, affirmative, counterfactual, sufficiency, necessity questions, and *et cetera*.

[6]The *reason* for $f(x)$ constitutes concepts stored in $\mathcal{S}$, accompanied by prototypes and their corresponding weights $\mathcal{W}$ for the class $y$. It can define how *literals* are structured, such as causal relationships, dependencies, or semantic groupings Wang et al. (2023a); Liu et al. (2023b).

## 2.2 FORMAL PRINCIPLES AND CONCEPTUAL BASIS

A *concept explanation* for a class can be local, focusing on providing *reasons* for an individual *instance*, global, examining the complete collection of *reasons* for different *instances*, or ideally both, covering a *class explanation* Holzinger et al. (2022); Ade-Ibijola & Okonkwo (2023). A *class explanation* aims to understand the overall logic of CNN by outlining the various *reasons* it assigned a particular class, requiring class-specific insights Zheng et al. (2017) and *axiomatic foundations*. We begin with the orthogonality of the first principles of CNN behaviour and explanation:

- Principle of Human Alignment: *Concept explanations* must be interpretable in human-understandable terms and cannot exist outside the interpretable space. Let $\iota : \mathcal{S} \to \mathcal{H}$ be the interpretability mapping, where $\mathcal{H}$ is the space of human-recognizable concepts. Then $\forall s \in \hat{f}(x), \iota(s) \neq \emptyset$.

- Principle of Causality: *Concept explanations* must reflect causal factors of the predictions, not spurious correlations. Let $do(\cdot)$ denote an intervention in Pearl's do-calculus Correa & Bareinboim. Then, for an *instance* $x$ and associating set of concepts $\mathcal{S} = \hat{f}(x)$: $\forall s \in \mathcal{S}, \quad P(f(x) \mid do(s = 0)) \neq P(f(x))$. That is, intervening in the presence of a concept must alter the output distribution, ensuring causal relevance.

- Principle of Consistency: *Concept explanations* for similar inputs must be stable and reproducible. Let $d_{\mathcal{X}} : \mathcal{X} \times \mathcal{X} \to \mathbb{R}^+$ be a metric on the input space, and $d_{\mathcal{S}} : \mathcal{S} \times \mathcal{S} \to \mathbb{R}^+$ a metric in the concept space. Then: $\forall x, x' \in \mathcal{X}, \quad d_{\mathcal{X}}(x, x') \leq \epsilon \implies d_{\mathcal{S}}\left(\hat{f}(x), \hat{f}(x')\right) \leq \delta$, for small $\epsilon, \delta > 0$. This ensures that small perturbations in inputs yield bounded changes in the explanations.

- Principle of Faithfulness: *Concept explanations* must preserve CNN decision logic. Predictions reconstructed via explanation concepts must match the CNN's predictions. For an explanation $\hat{f}(\cdot)$ and a prediction function $f(\cdot)$: $\forall x \in \mathcal{X}, \quad f(x) = f'(x \mid \hat{f}(x))$, where $f'(\cdot)$ is the surrogate function reconstructed from the explanations.

- Principle of Representation: Explanations must relate to CNN internal representations. Let $E(\cdot) : \mathcal{X} \to \mathbb{R}^k$ be the CNN representation function up to a latent layer. Then for any explanation $\hat{f}(x), \forall s \in \hat{f}(x), \exists g_s : \mathbb{R}^k \to \{0, 1\}$ such that $g_s(E(x)) = 1$.

## 2.3 FORMALIZATION OF AXIOMS

Given $\hat{f}(\cdot)$, $x$, and $f(x)$, for a *class question* $\boldsymbol{Q}$, let $\hat{f}(\boldsymbol{Q})$ be the set of literals $L$ explaining $f(x)$ while $L^T$ is the set of all *reasonable* ground-truth *laterals* and $f_{\boldsymbol{R}}(\cdot)$ is a *reasonable* concept extraction function from a given set of *literals*.

**Lemma 1.** $\boldsymbol{Q}$ *satisfies minimal sufficiency if the minimal set of reasonable literals* $f_{\boldsymbol{R}}(L)$ *provides sufficient explanations for all reasonable predictions in the set of ground truths* $f_{\boldsymbol{R}}(L^T)$. *Formally,* $f_{\boldsymbol{R}}(L) \equiv f_{\boldsymbol{R}}(L^T)$.

**Theorem 1.** *There exists a property exhibited by a concept explanation (in this case, we assume the concept importance $\mathcal{W}$), ensuring that the total importance of the explanation remains nearly unchanged such that:* $\sum_{l \in L} \mathcal{W}(l) \approx \sum_{l \in L^T} \mathcal{W}(l)$, *where* $|L| \ll |L^T|$.

The *axiomatic foundations* of $\hat{f}(\cdot)$ constituting the set of axioms $(\boldsymbol{g_1} \ldots, \boldsymbol{g_n}) \in \boldsymbol{\mathcal{G}^n}$, assert that the *concept explanations* are minimal, intuitive, and non-trivial without changing $\mathcal{W}$. We abstract formal properties that *concept explanations* must exhibit and define axioms as logical necessities that explanation methods must satisfy, as direct consequences of the formalized principles:

- Axiom 1 (*Interpretability $\boldsymbol{g_1} \in \{True, False\}$*): Derived from Principle of Human Alignment, $\boldsymbol{g_1} = True$ if $\forall L \in \hat{f}(\boldsymbol{Q}), L \neq \emptyset$.

- Axiom 2 *Relevance* ($\boldsymbol{g_2} \in \{0, 1\}$): Derived from the principle of Causality $\boldsymbol{g_2} = 1$ if $\forall L \in \hat{f}(\boldsymbol{Q}), \forall l \in L, l \models y$.

- Axiom 3 *Coherence* ($\boldsymbol{g_3} \in \{0, 1\}$): Derived from the principle of Consistency $\boldsymbol{g_3} = 1$ if $\forall L \in \hat{f}(\boldsymbol{Q})$ explaining $f(x), L \notin \hat{f}(\boldsymbol{Q}')$ for $x' \neq x$.

- Axiom 4 (*Fidelity $g_4 \in \{0, 1\}$*): Derived from the principle of Faithfulness, $g_4 = 1$ if $\forall L \in \hat{f}(\boldsymbol{Q})$, $\hat{f}(x) \equiv f(x)$, i.e., $\forall x \in X$, $\hat{f}(x)$ must preserve the predictive behaviour of $f(x)$ under concept perturbations.

- Axiom 5 (*Sanity $g_5 \in$ True, False, ⋆*): Derived from the principle of Representation, for $\boldsymbol{Q} = \langle \boldsymbol{T}, C, y \rangle$ and $\boldsymbol{Q}' = \langle \boldsymbol{T}', C', y' \rangle$ with $\boldsymbol{T} \neq \boldsymbol{T}'$, $C = C'$, and $y = y'$, $\forall L \in \hat{f}(\boldsymbol{Q})$, $\forall L' \in \hat{f}(\boldsymbol{Q}')$, $L \cap L' = \emptyset$ implies $\hat{f}(x) \neq \hat{f}(x')$. If $\boldsymbol{Q} = \boldsymbol{Q}'$, then $g_5 = \star$ (indeterminate).

## 2.4 SIGNIFICANCE OF THE PROPOSED AXIOMS

The axioms defined above and formalized comprehensively in the **Preliminaries** section of the appendix provide formal guarantees that *class explanations* are trustworthy. In the *dog* example from Figure 1, a robust and meaningful *concept explanation* should highlight a few *interpretable* concepts, such as *Ear*, *Eye*, *Nose*, and *Cheek*, which clearly support the *dog* classification. This aligns with the principles of Human Alignment on the one hand and Representation on the other hand, supporting the *Simplicity* axiom and avoiding excessive or misleading details such as the **Background**, and ensuring *coreness* by focusing on essential elements.

To maintain the principle of Consistency, the explanation must exclude contradictory concepts such as **Round Pupils** and **Slit Pupils**, which are typically associated with a *cat*. Furthermore, the principles of Representation and Consistency demand that concepts like *Cheek* generalize across multiple *dog* instances, not just a single example. These concepts should also be reflected in the associated *prototypes*. Crucially, the explanation must reflect the principle of Causality, where removing a causal *literal* significantly alters the CNN decision. This protects against spurious correlations and ensures that the explanation is grounded in the actual decision-making process of the model. Lastly, grounded in the principles of Consistency and Representation, the *Sanity* axiom requires that minor changes, such as image rotation, do not drastically affect the explanation, thus preserving its reliability in real-world scenarios.

## 3 EXPERIMENTS

### 3.1 CROSS-MODEL INVESTIGATIONS

We conducted cross-model investigations of two state-of-the-art (SOTA) concept explainers, CRAFT Fel et al. (2023) and ICE Zhang et al. (2021), each explaining Pytorch's pre-trained ResNet He et al. (2016) and Inception Szegedy et al. (2016) models using the ILSVRC2012 dataset Deng et al. (2009). We chose $c' = 32$ as recommended by Ramaswamy et al. (2023) and probed the penultimate CNN layers: $layer4$ for ResNet50 and $Mixed\_7c$ for Inceptionv3. Experiments were conducted on an NVIDIA GeForce RTX 4090. Since the CNNs (ResNet50 and Inceptionv3) were pre-trained, computational demands were modest. Both models achieved over 96% classification accuracy ($Acc$) for the target classes.

Figure 4 presents a cross-model analysis of an instance containing two breeds of dogs and a cassette player, answering class questions $\boldsymbol{Q}_{1-3}$. Each *class explanation* comprises 32 *concept explanations* grounded in axioms $g_1 - g_5$, although only the top three local concepts are shown for clarity. These are identified by unique **ID**s, *prototypes*, and $\mathcal{W}$ scores, ensuring trust through validation. This setup enables a detailed understanding of the concepts' contributions to CNN decisions (see the appendix for more results).

Although certain class questions are axiom-specific (e.g., Figure 1), the questions in Figure 4 require the application of all axioms, providing a comprehensive foundation. The figure illustrates how the axioms collectively assess the faithfulness of the explainer. Specifically, $g_1$ evaluates the alignment of the *concept explanations* with human understanding, while $g_2$ examines the causal relevance of the *concept explanations* to the CNN decision. The axiom $g_3$ quantifies the consistency of the *concept explanations* and $g_4$ measures the degree of faithfulness of the explainer. High scores in these dimensions serve to validate the quality of the explanations. Finally, $g_5$ evaluates robustness in terms of consistency and human alignment, returning Boolean outcomes for pairwise instance comparisons or ⋆ (indeterminate) for single instance evaluations. Figure 4 also highlights a fair[7]

---

[7]The comparison addresses $\boldsymbol{Q}_{1-3}$ for the specific instance and does not generalize to all class questions.

$Q_{1-3}$: *What concepts explain CNN decisions for Labrador Retriever, Yorkshire Terrier, and cassette player?*

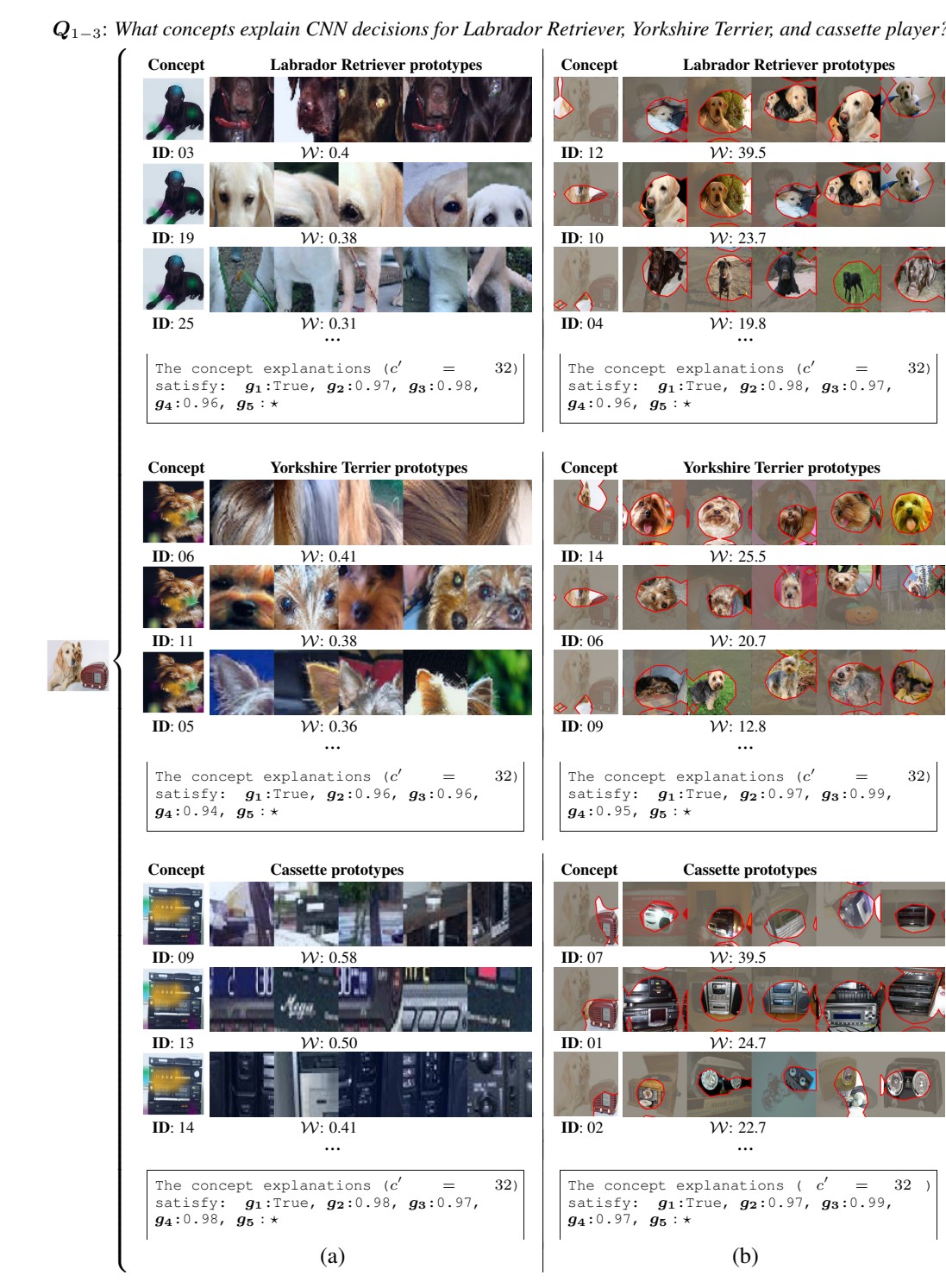

Figure 4: The three most significant local *concept explanations* (out of 32 local *concept explanations*) for an *instance* featuring a Labrador Retriever, a Yorkshire Terrier, and a cassette player, evaluated across their axiomatic foundations ($g_1, \dots, g_5$) for three *class questions*, $Q_1$, $Q_2$, and $Q_3$. (a) CRAFT Fel et al. (2023) on ResNet50 (colour maps represent concepts) and (b) ICE Zhang et al. (2021) on Inceptionv3 (red-bounded pixels represent concepts). Each explanation includes a unique ID, five prototypes, and a $\mathcal{W}$ score sorted in descending order by $\mathcal{W}$ scores. The proposed axiomatic foundations are shown in rectangular boxes below each class explanation. (*Best viewed in colour*).

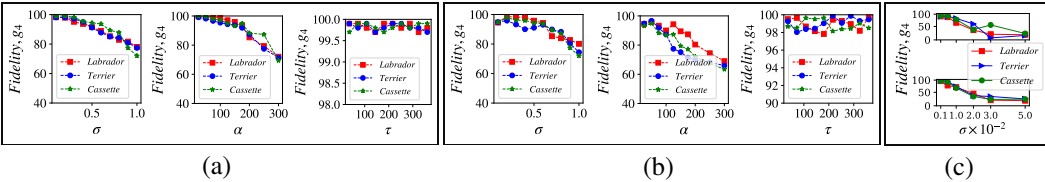

(a)             (b)             (c)

Figure 5: Impact of adversarial attacks on *Fidelity* $g_4$ (a) Frontdoor attacks: ICE Zhang et al. (2021) on ResNet50, (b) Frontdoor attacks: CRAFT Fel et al. (2023) on Inceptionv3, (c) Poisoning attacks: ICE Zhang et al. (2021) on ResNet50 (**Top**) and CRAFT Fel et al. (2023) on Inceptionv3 (**Bottom**). Similar trends were observed for other axioms under the adversarial conditions, with $True$ results for the Boolean axioms: $g_1$ and $g_5$.

cross-model comparison: CRAFT slightly outperforms ICE in *Fidelity* and *Relevance* while both methods maintain high *Coherence* and consistent adherence to the axiomatic framework.

## 3.2 ADVERSARIAL PERFORMANCE

We performed *Sanity* checks on the explainers under various adversarial conditions, including Frontdoor and Poisoning attacks (Figure 5). The parameters $\mu$, $\sigma$, $\lambda$, $\alpha$, and $\tau$ represent mean, standard deviation, smoothness, displacement, and rotation, respectively. For each condition, we generated concept explanations and evaluated classification accuracy ($Acc$) and axioms $g_n$ (see Figure 6 for a visual example).

The axiom $g_{10}$ was evaluated by comparing the *fidelity* $g_2$ between ideal and adversarial settings. As shown in Figure 6 (a, b), similar concepts produce different *prototypes* with reduced $Acc$ and $\mathcal{W}$ scores. Gaussian noise and warping significantly impact $Acc$ and *Fidelity* due to alterations in the value of the pixels ($\sigma$, $\alpha$), while rotation ($\tau$), which only shifts the position of the pixels, has a minimal effect (Figure 6 (c)). Figure 5 confirms that *Fidelity* is sensitive to changes in the value of pixels but resistant to spatial transformations. Our comprehensive evaluation of *Sanity* $g_5$ using other axioms validates *Sanity* of explainers in adversarial settings, highlighting the need for robustness checks during explainer development.

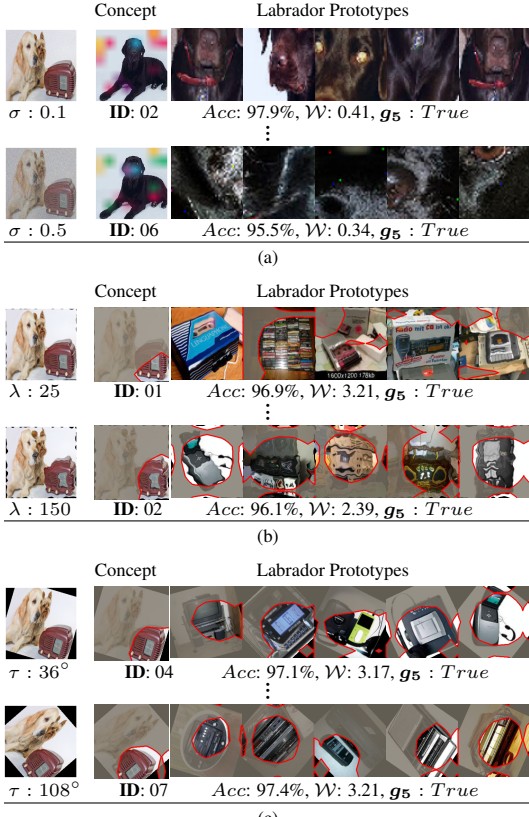

Figure 6: Explainer performance on ResNet50 under adversarial conditions for Labrador Retriever (a) Gaussian noise on CRAFT Fel et al. (2023), (b) warping on ICE Zhang et al. (2021), and (c) rotation on ICE Zhang et al. (2021).

## 4 DISCUSSION

### 4.1 IMPLICATIONS OF PROPOSED AXIOMS

Faithful *concept explanations* must successfully reflect the CNN classification logic to ensure *Relevance* Holzinger et al. (2022); Amgoud & Ben-Naim (2022). High *Coherence* and *Relevance* scores in Figure 4 indicate logical consistency and pertinence. *Representativity* supports generalization in similar instances, while *Agreement* links concepts to *prototypes*. *Fidelity* reflects the precision of both relevant and irrelevant concepts Ghorbani et al. (2019); Fel et al. (2023); Zhang et al. (2021). Lower Causality scores in Figure 4 suggest a nuanced influence of concepts on CNN decisions, enhancing trustworthiness.

Adversarial conditions can degrade the quality of the explanation, so a rational explanation must reflect this sensitivity Fel et al. (2023); Ade-Ibijola & Okonkwo (2023); Bao et al. (2023). *Concept explanations* should reveal causal, not simply correlated, *Reasons* Akpudo & Jang-Wook (2020); Wang et al. (2023a); Liu et al. (2023b); Ning et al. (2023). Although they are expected to be accurate, they should be *Relevant* to the *class explanation* Garcia & Johnson (2023), consistent with representative features of the image class Makridakis (1993); Wang et al. (2023b); Akpudo et al. (2024); Aronhime et al. (2014). For example, in Figure 1, concepts like *Ear*, *Eye*, *Nose*, and *Cheek* are relevant for a dog class, while accurate but irrelevant concepts (e.g., **Background**) may not contribute meaningfully.

## 4.2 INTERCONNECTEDNESS OF PROPOSED AXIOMS WITH DESIDERATA

The proposed *axiomatic foundations* imply and reinforce several other established desiderata, highlighting their interdependence. For instance, the *Sanity* axiom inherently requires *Ablation tests* to assess *Reliability* and *Stability*, while *Coherence* depends on analyzing the *Sparsity* of concept logits to ensure focused and interpretable explanations. These relationships suggest that axioms should be viewed not in isolation but as part of a structured framework, where validating one may support or necessitate the evaluation of others. This interconnectedness strengthens the overall robustness and trustworthiness of concept-based explanations in CNNs. Some axioms, like *Sanity*, may require inter-class evaluations. For example, the question in Figure 1: *How coherent is the **Cheek** concept for a dog breed (e.g. Bulldog) compared to those for another breed (e.g., Chihuahua)?* may necessitate new axioms. Similarly, in Figure 4, similar *prototypes* retrieved for different concepts (e.g., ICE concepts 12 and 10 for Labrador Retriever) suggest minimal plausibility differences. Although individually valid, merging them may better satisfy the *simplicity* axiom. Addressing such challenges may require interdisciplinary approaches beyond this study.

Finally, evaluating *Exhaustivity* is impractical due to subjectivity and uncertainty. Instead, emphasizing *Coreness*, as we have achieved, ensures faithful, trustworthy, and ethically sound explanations with broader impact (see the **Broader Impact** of the Appendix for a detailed discussion).

## 5 RELATED WORKS

In line with the larger expectation that AI should assist rather than replace humans Marques-Silva & Ignatiev (2022), human-in-the-loop frameworks have become essential to build trust and accountability in AI systems Atienza et al. (2024); Ghosh (2023). CNNs, while not inherently interpretable, support concept extraction through their layered filtering operations, allowing classification and post hoc interpretation Kim & Chae (2024); Akpudo et al. (2025b). However, poorly performing CNNs, particularly under adversarial conditions, can increase interclass similarity, generate meaningless concepts, and doubt the sanity of the explainer Karimi et al. (2020); Bao et al. (2023).

Traditional attribution-based methods, such as GradCAM, highlight influential input regions, offering insight into where a model looked, but not what it saw Ribeiro et al. (2016); Liu et al. (2023b); Amgoud & Ben-Naim (2022); Garcia & Johnson (2023). These methods do not satisfy key axiomatic foundations, such as *Causality*, require expert judgment, and are vulnerable to adversarial attacks Salahuddin et al. (2022); Preechakul et al. (2022); Chakraborty et al. (2022). Concept-based approaches offer a more abstract and cognitively aligned perspective Kim et al. (2018); Zhang et al. (2021); Ghorbani et al. (2019); Fel et al. (2023). Still, supervised methods relying on predefined concepts face significant limitations: they often fail to capture nuanced features Kim et al. (2018); Ramaswamy et al. (2023), are harder to learn than class labels Ramaswamy et al. (2023), and can introduce bias Fel et al. (2023); Akpudo et al. (2025b). These challenges have catalyzed the development of unsupervised concept-based methods centered on automatic concept discovery Ghorbani et al. (2019); Zhang et al. (2021); Fel et al. (2023); Akpudo et al. (2024); Kim et al. (2018); Ramaswamy et al. (2023); Akpudo et al. (2025a).

Recent frameworks integrate dimensionality reduction techniques, that is, reducers in CNNs, to discover concepts without manual labelling Akpudo et al. (2025a; 2024; 2023); Zhang et al. (2021); Fel et al. (2023), significantly improving explainability and interpretive diversity Gupta & Narayanan (2024); Weber et al. (2023). ACE Ghorbani et al. (2019) pioneered this direction, segmenting images, grouping similar regions, and rejecting outliers to define concepts. However, its outlier rejec-

tion can lead to information loss, and concept importance may vary across instances. ICE Zhang et al. (2021) and CRAFT Fel et al. (2023) build on this by using NMF; ICE shows strong performance, while CRAFT introduces recursive decomposition for refined explanations. Generative approaches Akpudo et al. (2025a); Brocki & Chung (2019); Liu et al. (2023a) offer further flexibility, but require extensive tuning and introduce interpretability challenges Takeishi & Kawahara (2020), including misalignment between model metrics and human perception Zhang et al. (2016).

Despite their promise, unsupervised explainers lack a formal axiomatic foundation. This makes cross-method comparisons difficult due to inconsistent assumptions, domain-specific constraints, and the absence of standardized evaluation criteria Akpudo et al. (2024). The inability to validate automatically discovered concepts further undermines the stakeholders' trust Akpudo et al. (2025a); Zhang et al. (2021); Ghorbani et al. (2019); Fel et al. (2023); Akpudo et al. (2024). Many explainers still rely on cherry-picked examples that appear intuitive but are anecdotal and unsupported by rigorous evaluation He et al. (2025); Nauta et al. (2023b); Akpudo et al. (2025b), allowing anecdotal inspection to dominate and weakening the credibility of XAI research Upadhyay et al. (2025); He et al. (2025). To address these gaps, we ask: *What axiomatic foundations guarantee faithful and trustworthy concept explanations?* We argue that concept explanations must satisfy key formal properties to be considered trustworthy Amgoud & Ben-Naim (2022); Sundararajan et al. (2017); Chen et al. (2023). As illustrated in Figure 1, our work introduces the notion of coreness and emphasizes cognitive alignment and ethical considerations Akpudo et al. (2025b); Lincoln IV (2025); Upadhyay et al. (2025). For the first time, we formally investigate the axiomatic properties of unsupervised concept-based explainers and propose a framework for harmonized evaluation and cross-model comparison. Rather than debating the superiority of supervised versus unsupervised concepts, a question already settled in the literature Kim et al. (2018); Zhang et al. (2021); Fel et al. (2023); Ghorbani et al. (2019); Kori et al. (2023); Nauta et al. (2023a); Kim & Chae (2024), we define a set of desirable axioms that any concept explanation should satisfy to earn stakeholder trust. Our framework builds on advances in concept discovery, prototypical representation, and abstraction, offering a principled foundation for evaluating the faithfulness of concept-based explanations Akpudo et al. (2025a; 2024); Sundararajan et al. (2017).

## 6 ETHICAL STATEMENT

The paper is foundational and theoretical, and does not present or implement technology with direct societal deployment. While the work does not involve crowdsourcing or research with human subjects, nor pose immediate risks, it could influence how future systems align with human values, necessitating the protection of CNNs and explainers from adversarial conditions. The proposed axiomatic foundations safeguard explainers by demanding that faithful and trustworthy explanations adhere to desired properties, even under adversarial conditions. Poor performance under such conditions can doubt the explainer's sanity, highlighting a potential flaw detected through axiomatic testing. The paper ensures reproducibility through open datasets, pretrained models, detailed methodology, and clear experimental results with supporting visuals and metrics.

## 7 CONCLUSIONS AND FUTURE WORKS

This study introduces a novel framework for evaluating concept-based explanations of CNNs through a unified axiomatic foundation, emphasizing trustworthiness. Key contributions include defining these axioms, conducting quantitative investigations to assess explainer performance under adversarial conditions, and offering insights for developing more transparent AI systems. Experiments with the ILSVRC2012 dataset highlight the critical role of robust axiomatic foundations, particularly in adversarial conditions. The findings emphasize the importance of rigorous testing to ensure the reliability and practical value of explainers. This work provides foundations for deploying resilient explainers and underscores the need for future research to enhance their robustness and ensure trust. The axiomatic foundations proposed for CNN explainers are model-agnostic and could extend to transformers, despite architectural differences. While CNNs extract local features, Vision Transformers (ViTs) capture global dependencies through self-attention, requiring a redefinition of concept discovery. However, the proposed axiomatic foundations remain essential for ensuring trustworthy explanations. Applying these principles to ViTs and multi-modal models presents an opportunity to advance interpretability across SOTA and custom visual categorization models.

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
