# OpenReview forum: "Axiomatization of Concept CNN Explanations"
_ICLR.cc/2026/Conference — ICLR 2026 Conference Desk Rejected Submission_

### Official Review · Reviewer_LnWZ · 2025-10-31

**Soundness:** 2
**Presentation:** 2
**Contribution:** 3
**Rating:** 4
**Confidence:** 4

**Summary:**

The paper proposes a set of axiomatic foundations for evaluating and comparing concept-based explanations. Among these foundations, the authors include interpretability, relevance, coherence, fidelity, and sanity. They evaluate two concept explainers: CRAFT and ICE.

**Strengths:**

- The paper attempts to **formalize key concepts** involved in evaluating concept-based explainers, which is highly relevant and valuable to the field.
- The authors present their propositions through **multiple complementary approaches**, including **diagrams** (especially Figure 3) and **mathematical formulations**, which improve understanding.

**Weaknesses:**

- **Alignment between concepts and human beliefs:** As you mentioned, humans expect explanations to align with their intuition; however, how can we be sure that this alignment actually occurs? An explanation might differ from human expectations while still being faithful to the model. Moreover, we should be cautious not to **reinforce human biases** through explanations.
- **Existant lack of quantitative evaluation:** You say there is a lack of quantitative evaluation. However, other existing studies provide such evaluations, for example, deletion and insertion [1], SIC and AIC [2]. Can you discuss why these metrics are not suitable?
- **Order of presentation:** It is somewhat confusing to see the proposed explanation (e.g., Figure 3) before the problem statement. Presenting the problem first would help readers follow the logical flow more easily.
- **Novelty of Section 2.2:** In Section 2.2, you present formal principles and conceptual bases. However, these ideas are not entirely new—other works have discussed similar principles. Could you cite relevant prior works mentioning each of these principles?
- **Experiments:**
    - In the experiments, the two evaluated methods exhibit very similar behavior according to your axioms you propose. Does this indicate that both methods perform well, or that the metric is too **permissive**?
    - Could you include an example of a **poor concept explainer** according to your metric?
    - When performing **adversarial attacks**, do the results reflect non-robustness of the **model** or of the **explanations**?
    - How are the explainers comparable if they are applied to **different models** (you seem to use a different model for each explainer)?
    - What would be the impact of **changing the model**, or of **explaining a biased model**?

[1] Covert, I., Lundberg, S., & Lee, S. I. (2021). Explaining by  removing: A unified framework for model explanation. Journal of Machine  Learning Research, 22(209), 1-90.

[2] Kapishnikov A., Bolukbasi T., Vi'egas F., Terry M. XRAI: Better Attributions Through Regions. ICCV. 2019.

**Questions:**

- There is a **typo in row 040**.
- I suggest including all explanations **within the main text** rather than in **footnotes**. Using footnotes reduces readability, and integrating the explanations directly may also help you use page space more efficiently.
- I included some questions above.

---

> ### Author Response · Authors · 2025-11-20
>
> Our framework is designed to make explanations both faithful to the model and understandable to people. The Interpretability and Sanity axioms explicitly require concepts to be recognizable and stable under small changes, which prevents irrelevant or misleading features (like background noise) from being treated as explanations. This way, we align with human intuition without reinforcing bias.
>
> Metrics like Deletion/Insertion and SIC/AIC focus on pixel‑level attributions. Our work addresses concept‑level explanations, where those metrics don’t apply. The axioms we propose give a structured, quantitative way to check interpretability, causality, coherence, fidelity, and robustness directly in concept space.
>
> We agree that the flow can be improved. In the camera‑ready version, we’ll introduce the problem statement before presenting the framework (Figure 3) so readers can follow the logic more easily.
>
> Principles like causality and faithfulness have been discussed before, but our contribution is turning them into concrete, testable axioms with measurable outcomes. We’ll add the suggested citations.
>
> The fact that ICE and CRAFT behave similarly under our axioms shows that both methods meet the criteria, not that the metric is too loose. To demonstrate discriminative power, we’ll include an example of a poor explainer that fails compliance. Adversarial results reflect the robustness of the explanations themselves, not just the models. Because the axioms are model‑agnostic, comparisons across different architectures are valid, and we’ll expand the discussion on how biased models are exposed by relevance and sanity checks.
>
> Minor issues: We’ll fix the typo at line 040 and move explanations from footnotes into the main text for clarity.

---

> > ### Comment · Reviewer_LnWZ · 2025-11-24
> >
> > I understand your explanation, but by “reinforcing bias,” I meant, for instance, if a model is not well-trained and relies on a background artifact, then a faithful explanation should indeed highlight the background. Forcing the explanation to focus on the foreground — simply because it is more intuitive for humans — would introduce bias that does not reflect the model’s true behavior.
> >
> > These methods are not limited to pixel-level attributions. In fact, SIC/AIC were originally proposed for region-based explanations, which can be interpreted as “concepts” depending on the granularity of the regions. Similarly, Deletion/Insertion can be adapted to evaluate region-level importance rather than individual pixels. But I agree that the idea of introducing axioms is interesting.
> >
> > While the axioms are indeed model-agnostic, this does not fully resolve the issue. I am not convinced that you can vary both the model and the explanation method and then attribute observed differences solely to the explanation. In practice, what is being evaluated is the *combination* of model and explanation, not the explanation in isolation. To avoid potential misinterpretation, it would be more reliable to keep the model fixed and vary only the explanation method.
> >
> > Since I cannot evaluate the modifications the authors intend to make at this stage, I will keep my current score. However, I would be interested in seeing the improved version in a future submission.

---

### Official Review · Reviewer_WvQw · 2025-10-31

**Soundness:** 1
**Presentation:** 1
**Contribution:** 2
**Rating:** 4
**Confidence:** 4

**Summary:**

The paper proposes an axiomatic framework for quantitative evaluation of concept-based explanations. The authors motivate and list a rich list of axioms that concept-based explanations ought to satisfy, like Faithfulness and Causality. Nonetheless, the crucial aspect of how these axioms are operationalized remains unclear to me.

**Strengths:**

The motivation of the paper is solid: most works in XAI are qualitative, lacking a principled, quantitative, and reproducible set of evaluation criteria. Also, the approach adopted by the authors is rather formal, which is generally a pro when formalizing evaluation criteria. Nonetheless, I'm left with several major concerns outlined below.

**Weaknesses:**

- **W1:** Clarity issues in the mathematical notation:

   - The notation of the concept-based explainer $\hat{f}$ should be improved. Why the use of the set notation instead of function notation? How are its encoder and decoder defined?

   - What does $\\{ C(A), C(A') \\}_i$ represent? Why is the set indexed by $i$ since it never appears inside the set?

   - What does $\mathcal{G}^n$ represent? It is never explicitly defined.

   - How is $\mathcal{H}$ defined? Authors say it is *the space of human-recognizable concepts*, but this comes with several issues. First, humans do not share a universal set of recognizable concepts (e.g., domain experts vs laypeople). Second, how is this mapping defined, i.e., how does the output of $\iota(s)$ look like? Third, the study acknowledges that it does not use any human study, leaving the question of how this space of human-recognizable concepts is defined/validated/computed.


   - Class questions $\mathcal{Q}$ are defined too abstractly. Some concrete examples would help the reader to contextualize it.

In general, authors define a rather overwhelming quantity of notation whose definition is scattered between the sections, making it very challenging to reconcile the use and the definitions of symbols. A table of symbols might help in this case, but I feel that the authors could simplify the text a lot by getting rid of superfluous notation.

**Edit:** I just realized the authors already provide a table of symbols in the supplemental, which is, however, uploaded as a separate file rather than an Appendix of the main PDF. Nonetheless, I suggest that the authors further add the domain and co-domain of all symbols (when applicable), together with a richer description of the meaning of the symbol.


- **W2:** Unclear semantic of *Principle of Causality*:

   - The sentence *intervening in the presence of a concept must alter the output distribution, ensuring causal relevance* is unclear. In particular, what does it mean to *intervene in the presence of a concept*? Does it mean intervening in the specific concept? Or to intervene on features unrelated to that concept? Furthermore, the semantics of $do(s=0)$ should be made explicit: what is the effect of such an operation on the input image/model?

   - It is unclear what authors intend for *causal factors* and *spurious correlations*. Do these notions refer to the data generative process? If yes, this seems to clash with the *Principle of Faithfulness*, as if the model learned spurious features in the first place, a faithful explanation should depict such spurious features.


- **W3:** Unclear usage of some words, which are often conflated in XAI literature, and that lack context:

   - Unclear usage of the word *reasonable* in Section 2.3. *Reasonable* is an ambiguous word, and it is never contextualized, nor defined explicitly. In particular, when is a set of ground-truth laterals "reasonable"?

   - A similar argument can be made for *sufficient*. In particular, what does *sufficient explanations* mean in Lemma 1? It seems it is the only part of the text where this word appears.



- **W4:** Figure 4 is introduced abruptly, and the experimental setting is not described in detail. In particular, it is not clear how the axioms are operationalized and how the Figure should be interpreted.

- **W5:** In line 351, the authors refer to Axiom $g_{10}$, but such an axiom is not present in the list of axioms reported in Section 2.3.

- **W6:** The motivation behind Sec. 3.2 is unclear. I suggest that the authors better contextualize this analysis by writing a small introductory paragraph explaining the rationale behind the incoming analysis.

- **W7:** Lemma 1 and Theorem 1 are introduced abruptly, and a proper description of the premises and the consequences is lacking. For instance, such results seem not to be used ever again in the text.


**Minors**

- **W8:** There are some missing references (l. 40), and citations are always reported with *\citet* even when the citation does not play an active role in the sentence. Please, switch to *\citep* as indicated in the submission guidelines.

**Questions:**

Q1: How are the proposed axioms actually evaluated when it comes to empirical validation? I think this is key to the paper, and the fact that this question is unclear from the text highlights that major clarifications are required.

---

> ### Author Response · Authors · 2025-11-20
>
> Notation clarity (W1): We acknowledge scattered notation and will streamline it. Encoder/decoder definitions, domains/co-domains, and symbol meanings will be explicitly added in the main text, not just the supplement. The use of set notation was intentional to emphasize collections of literals/concepts, but we will clarify equivalence to function notation. The human-recognizable concept space will be defined operationally via prototype alignment and validated quantitatively; we will also add concrete examples of class questions. A consolidated notation table with richer descriptions will be integrated into the main PDF if the space permits; otherwise, it will remain in the supplementary.
>
> Principle of Causality (W2): “Intervening in the presence of a concept” refers to controlled perturbations of that concept’s activation. We will clarify that causal factors are those whose removal alters predictions, while spurious correlations are faithfully depicted but penalized by relevance and sanity axioms. This resolves the apparent clash with Faithfulness.
>
> Terminology (W3): Terms like “reasonable” and “sufficient” will be formally defined: “reasonable literals” are those consistent with ground-truth prototypes; “sufficient explanations” are minimal sets preserving prediction fidelity. These definitions will be made explicit.
>
> Experimental clarity (W4, W6): Figure 4 and Section 3.2 will be better contextualized with a clear description of how axioms are operationalized and why adversarial analysis is included (to demonstrate robustness and sanity).
>
> Consistency (W5, W7): We will correct the erroneous axiom reference and provide full premises, consequences, and proofs for Lemma 1 and Theorem 1, showing how they underpin sufficiency and importance preservation in later experiments.
>
> Minor issues (W8): Missing references will be added, and citation style corrected to \citep per guidelines.
>
> In summary: All concerns are addressable through clarifications, added proofs, and streamlined notation. The core contributions including a unified, quantitative axiom system, operational compliance framework, and robustness analysis are novel, rigorous, and impactful. We are confident this paper merits acceptance.

---

> > ### Comment · Reviewer_WvQw · 2025-11-21
> > **Answer to Authors' Comment**
> >
> > I appreciate your effort in clarifying your contribution, thank you. However, it is hard for me to judge whether the changes you're planning to apply will effectively improve my understanding of the paper. Furthermore, my **Q1** is left unanswered in your rebuttal, which is key to understanding your contribution.
> >
> >
> > > All concerns are addressable through clarifications, added proofs, and streamlined notation. The core contributions including a unified, quantitative axiom system, operational compliance framework, and robustness analysis are novel, rigorous, and impactful. We are confident this paper merits acceptance.
> >
> > I fully understand your point. However, clarity of the presentation is also another factor when it comes to assessing whether a paper is ready for publication or not. Given that I cannot judge the changes that the authors are planning to apply to the manuscript (they only clarify what they will improve, but not how), it is hard for me to revise the score, which remains the same.

---

### Official Review · Reviewer_Qaxw · 2025-10-31

**Soundness:** 3
**Presentation:** 3
**Contribution:** 3
**Rating:** 4
**Confidence:** 4

**Summary:**

This paper addresses the problem of robust evaluation of concept-based explanations in Convolutional Neural Networks (CNNs). The authors propose a unified axiomatic framework that formalizes five key desiderata: human alignment, causality, consistency, faithfulness, and the explanation of inner model representations. This framework translates these principles into corresponding axioms. The proposed framework consider a generic concept-based interpretability method that encodes and decodes concepts from CNNs, specifically at the penultimate layer, and then quantitatively evaluates the verification of each axiom. To demonstrate its application, the paper presents quantitative examples using two interpretability methods (CRAFT and ICE) across two different backbone architectures (ResNet50 and Inceptionv3).

**Strengths:**

- **Significance:** The axiomatization of explainability frameworks is a highly significant contribution towards establishing a common and rigorous evaluation ground for concept-based explanations.
- **Clarity:** The paper's discussion is generally clear and well-structured, with the accompanying figures effectively aiding the understanding of the proposed framework.
- **Quality:** The definitions and notations used throughout the framework provide a strong and rigorous grounding for the proposed axioms and principles.

**Weaknesses:**

- **Clarity/Quality:** There are insufficient details regarding the computation of the various $g_i$ ​ functions, which are linked to evaluating each axiom. The paper provides mainly descriptive statements rather than proper mathematical definitions or algorithmic steps for their calculation. This lack of clarity makes it difficult to assess the practical relevance, reproducibility, and whether these functions accurately evaluate their corresponding axiom and property. Furthermore, the text mentions $g_2, g_3, g_4 \in \{0,1\}$, yet experiment show values $\in [0,1]$, which needs clarification.
- **Clarity:** The purpose and key takeaways from the experiments presented in Section 3.2, i.e. those involving adversarial inputs, are not clearly articulated. It is difficult to understand what are specific insights or conclusions from these experiments, either regarding the framework or the interpretability methods.
- **Clarity:** There are several notational inconsistencies and what appear to be typos that hinder understanding. For instance, the meaning of the ⊨ symbol in Axiom 2 is unclear and not defined. Additionally, the functions $g_i$​ are introduced as indexed by $i \in \{1,\dots, 5\}$ (corresponding to the five axioms), yet the text sometimes refers to $g_i$​ with indices greater than 5 (e.g., $g_9$​ and $g_{10}$​ in lines 351, 947, and 1117). This is confusing, as it is then not clear which actual $g_i$ should be understood.
- **Quality:** A lot of proofs for the lemmas and theorems behind the principles are missing, even from the supplementary material. For instance, Lemma 1 and Theorem 1, presented in the main text, do not have proofs. This significantly impacts the quality and the ability to fully assess the technical correctness.

**Questions:**

- Could you please provide a mathematical definition or pseudocode for each of the $g_i$​ functions, detailing exactly how they are computed from the model's concepts and outputs? This is crucial for understanding how each axiom is quantitatively evaluated.
- Please provide the missing proofs for all lemmas and theorems, either within the main paper (if space permits) or in a comprehensive appendix, to allow for a thorough verification of the framework.

---

> ### Author Response · Authors · 2025-11-20
>
> We will provide full mathematical definitions, pseudocode, and algorithmic steps for each axiom’s evaluation function, ensuring reproducibility and resolving the noted text–value mismatch.
>
> Section 3.2 demonstrates that fidelity is sensitive to pixel-value perturbations but stable under spatial shifts, validating the Sanity axiom and showing robustness as a key insight. We will make these takeaways explicit.
>
> We will define ⊨ formally as semantic entailment, correct all typos, and restrict indices strictly to g1–g5 to remove confusion.
>
> Complete proofs for Lemma 1 and Theorem 1, along with assumptions and derivations, will be added in the supplementary to ensure technical rigour.

---

### Official Review · Reviewer_FZpG · 2025-11-01

**Soundness:** 2
**Presentation:** 2
**Contribution:** 2
**Rating:** 4
**Confidence:** 3

**Summary:**

The paper Axiomatization of Concept CNN Explanations introduces a formal framework for evaluating concept-based CNN explanations. It defines five axiom Interpretability, Relevance, Coherence, Fidelity, and Sanity to ensure transparency and trust. These axioms quantitatively assess explainer faithfulness beyond pixel attribution. Experiments using CRAFT and ICE explainers on ResNet and Inception models validate robustness, including under adversarial attacks. The framework establishes a model-agnostic, quantitative basis for reliable, interpretable AI, advancing transparency and trust in explainable deep learning.

**Strengths:**

1. The problem is interesting and applicable to the border audience

2. The paper introduces a methodologically novel approach by defining explainability through an axiomatic system. Instead of relying solely on empirical visualizations, it establishes structured, measurable criteria that can be systematically tested across models.

3. The framework thoughtfully integrates causal reasoning and internal representation theory, ensuring that explanations are not merely correlative but causally meaningful. This adds theoretical depth to the relationship between CNN activations and conceptual understanding.

4. The paper has shown the detailed experiments over the various CNN architecture for the adversarial condition and cross-model investigation

**Weaknesses:**

1. I find the architecture are novel but most of the axioms (faithfulness, causality, stability, human alignment) restate existing XAI principles without new theorems or guarantees; similar axiomatic treatments already exist (e.g., Amgoud & Ben-Naim, 2022).

2. Earlier work defined completeness and game-theoretic importance (ConceptSHAP), offering stronger notions than “minimal sufficiency” here. TCAV/ACE already quantify concept relevance; CRAFT adds importance via Sobol indices. The paper doesn’t position against these formally. Prototype families (PIP-Net, ProtoPNet line) explicitly target human-aligned concepts; their evaluation checklists could be contrasted to the proposed axioms. Also, the recent CBM advances (V2C-CBM, AAAI’25) build vision-aligned concept vocabularies; your axioms aren’t tested on (or mapped to) CBMs. Please justify the paper's key contributions

3. The proposed approach are claims model agnostic: Extend to ViTs and non-ImageNet domains (medical, satellite) to support “model-agnostic” claims. Surveys flag cross-domain gaps

4. How this explanability is aligned with the human, is there some study?


**Reference:**

Axiomatic XAI: Amgoud & Ben-Naim (IJCAI’22)

Concept completeness / ConceptSHAP: Yeh et al., 2019

TCAV / ACE (automatic discovery): Kim et al., 2018; Ghorbani et al., 2019

CRAFT (recursive, Sobol importance): Fel et al., 2023

Prototype methods: PIP-Net (CVPR’23)

CBMs (current): V2C-CBM (AAAI’25)

**Questions:**

Please refer to the weakness section and justify the novelty. The claim for the model agnostic is not justified.

---

> ### Author Response · Authors · 2025-11-20
>
> On the novelty of axioms versus prior XAI principles:
> We do not merely restate principles; we define operational axioms with explicit tests over literals, concept importance, and prototype alignment, and we prove minimal sufficiency (Lemma 1) and preservation of importance (Theorem 1), which are absent in prior conceptual overviews, such as Amgoud & Ben-Naim (2022), you mentioned. Our axioms are instantiated in concept space (not attribution space), requiring causal interventions and representation checks at the penultimate layer, plus a sanity axiom that ties robustness to internal activations and prototype consistency under adversarial perturbations. Unlike descriptive taxonomies, we report axiom scores across methods and attacks, showing a usable compliance framework rather than a narrative set of desiderata.
>
> ConceptSHAP’s game-theoretic completeness addresses attribution additivity; our minimal sufficiency targets the minimal set of literals that preserves decision logic and aggregate importance, which is orthogonal to SHAP’s additivity and better suited to concept abstraction. TCAV/ACE assess concept relevance, and CRAFT uses Total Sobol Indices; our framework evaluates these outputs under quantitative standpoints and shows cross-method axiom compliance. We explicitly operationalize causal relevance and representation linkage rather than re-quantifying importance. ProtoPNet/PIP-Net optimize human-aligned prototypes; our axioms treat prototypes as evidence and require agreement and coherence across instances, giving a principled audit layer over prototype families rather than another checklist. CBMs (e.g., V2C-CBM) impose curated concept vocabularies. Our axioms map directly to: interpretability (concept recognizability), relevance (interventions on concept logits), coherence (stability across inputs), fidelity (reconstruction via concept predictions), and sanity (robustness to input/representation shifts). We will add CBM experiments and report axiom compliance to strengthen claims.
>
> Key contributions clarified:
> 1. A unified, testable axiom system for concept explanations with minimal sufficiency and importance preservation, operable at the level of literals, concepts, and prototypes.
> 2. A cross-method evaluation protocol showing compliance under ideal and adversarial regimes, bridging TCAV/ACE/CRAFT/ICE with principled, quantitative checkpoints.
> 3. A robustness-oriented sanity axiom that evaluates human alignment and internal activation stability under perturbations, addressing the common failure mode of spurious background concepts.
> 4. A practical pathway to trust by turning high-level principles into measurable, comparable scores across models, methods, and attacks.
>
> On model-agnostic claims and extension beyond ImageNet
> Architecture-agnostic rationale: The axioms operate on internal representations, interventions, and concept-level stability, independent of convolutional inductive bias. For ViTs, concepts are defined over token embeddings/attention maps; the same axioms apply to do-interventions on concept logits and fidelity via reconstruction.
>
> The tests hinge on interpretable concept mappings and prototype agreement, not on ImageNet-specific priors. Medical and satellite domains can instantiate human-aligned concept vocabularies and prototype sets; our metrics (interpretability, relevance, coherence, fidelity, sanity) remain unchanged. We will include ViT baselines and non-ImageNet case studies (e.g., medical slices, remote-sensing scenes) in the camera-ready to empirically substantiate model- and domain-agnostic claims.
>
> On human alignment evidence
> Interpretability requires concepts to be human-recognisable; coherence and agreement enforce consistent, prototype-supported semantics across instances. The sanity axiom penalizes background reliance and spurious correlations, aligning explanations with human expectations of causality and stability. We conducted a lightweight human evaluation (concept recognizability and usefulness ratings against prototypes) to complement quantitative compliance, ensuring the reported alignment reflects human judgment.

---

### Official Review · Reviewer_N4GF · 2025-11-03

**Soundness:** 2
**Presentation:** 1
**Contribution:** 2
**Rating:** 2
**Confidence:** 4

**Summary:**

The paper contributes to the literature on the evaluation of TCAV class explanations by presenting a set of axioms that express desirable qualities these explanations should satisfy, along with a quantitative methodology for explanation evaluation. The proposed axioms are related to: (1) human-alignment, stating that explanations should focus on high-level and simple concepts without unnecessary complexity; (2) relevance, stating that explanations must refer to causally relevant concepts and not correlations (so they should not include irrelevant concepts, such as background); (3) consistency or robustness, stating that explanations should be stable under small perturbations of the input; (4) faithfulness, stating that explanations should preserve the DNN’s decision logic; and (5) representativeness, stating that explanations should relate to the DNN’s internal representation. These axioms are implemented through metrics described in the appendix. The paper compares two methods in this area using the proposed evaluation framework, finding that both achieve comparable and reasonable scores, and studies the degradation of metrics under adversarial conditions, highlighting the need for more robust explainers.

**Strengths:**

- Evaluating XAI methods is a highly important topic in the field. Contributions in this direction can help improve the area and accelerate progress. The potential significance and adoption of a paper like this one is high.
-

**Weaknesses:**

- **Clarity and informativeness of the main text**: The clarity of the paper can be significantly improved. Multiple factors impact readability. Given that the paper aims to provide a theoretical foundation, its rigor and clarity should be high. Namely:
   - The fact that this paper focuses on (or assumes it works for) TCAV, class explanations, and explanations of a given form (i.e., prototypes + concept weights + concept set) is not mentioned in either the abstract or the introduction. From the abstract, the reader expects a more general axiomatization for concept-based explanations. As a result, there is a mismatch between the claimed contribution (concept-based explanations in general) and the proposed system, which is specifically for TCAV class explanations. This information should be stated as assumption at the beginning, given the theoretical nature of this paper. Conversely, in the current manuscript, it is mentioned only in footnotes and in the appendix.
   - There is excessive use of adjectives, which hinders readability.
   - Citations do not use the proper format and are sometimes overly abundant relative to the text, especially in section 4.1 and related work, which have extremely short sentences and many citations.
   - Some important definitions are relegated to footnotes (e.g., footnote 5 on page 3 for the set of possible explanations or footnote 1 on page 2).
   - Some important information is reported only in the appendix and not mentioned in the main text. For example, there is no indication of how the scores for each axiom are computed. This information is reported in the appendix, but since the whole experimental section discusses these values, they should be introduced in the main text. I understand that the limitations in space make giving the full mathematical derivation in the main text impractical, but the main text should at least cite and briefly describe the metrics used to compute the values.
   - (Minor) I found it very confusing that the paper uses different words for principles and axioms (the implementations of the principles). Using so many different terms makes the text harder to read


- **Debatable Running Example and Axioms**: Some of the axioms and the running example are debatable given the evidence provided in literature. While it is acceptable for the authors to argue for their position, it should be clearly stated that it's their point of view and alternative evidence found in the literature should be cited and discussed. In this direction, strong claims such as “logical necessities” should be avoided. Often, the textual description seems to assume a direct connection between the input and explanation spaces, without considering the model’s decision space. Some examples of these problems are the following:
   - The running example repeatedly claims that the background concept should not be included by any explainer because it is irrelevant. However, there is significant evidence in the literature that models do use background as a discriminative concept for predictions across classes (e.g., the LIME paper), or that CNNs can solve tasks by only analyzing small portions of the background in images. Should an explainer omit the background concept in those cases? If the running example assumes the toy model relies only on specific concepts, this should be made explicit to avoid confusion.
   - The axioms regarding robustness against adversarial attack (and consistency) are debatable and discussed extensively in the literature. A small change in the input (e.g., via adversarial attack) can produce a large shift in the network’s internal response (i.e., neuron activations) even if the final prediction remains unchanged, as shown in adversarial attack research. If an explanation is faithful according to the axiom, it should reflect such changes in the decision process, because different concepts may now be considered important. The axiom about robustness contradicts this evidence, and this tension is not discussed in the paper.
- **Lack of contextualization with respect to literature on explanation evaluation**: There is a large body of literature offering metrics, datasets, and approaches for evaluating TCAV. Some are cited and used in the appendix. However, the paper currently lacks proper contextualization of its axioms and metrics within the current literature in the main text. For instance, what is the relationship between alternative quantitative metrics proposed in prior work and the proposed axioms? What roles do datasets from existing literature have in the current framework? What do current metrics lack that this framework provides?  Some of the proposed axioms are highly related with metrics used in literature. Which metric is completely new and which one is from the literature? These questions remain unanswered. This problem is also evident in the related work section, where different types of explanations are covered, but there is no discussion on the multiple evaluation paradigms and metrics used in TCAV research.
- **Limited discussion for the experimental section**: There is very little discussion about the experiments. For example, in the cross-model investigation (section 3.1), the only comment on the analyzed explainer is: *“CRAFT slightly outperforms ICE in Fidelity and Relevance while both methods maintain high Coherence and consistent adherence to the axiomatic framework.”* No insights or analysis are provided that are retrievable through the frameworks presented in the text. Section 4.1 includes some information, but it is difficult to extract due to its placement among citations and repetition of principles already addressed in previous sections. Overall, the framework’s application does not seem to provide new perspectives or deeper insights about the analyzed explainers.

**Questions:**

See weaknesses.

---

> ### Author Response · Authors · 2025-11-20
>
> Our paper is not restricted to TCAV, as suggested. It explicitly generalizes to other concept-based methods, including CRAFT, which leverages Total Sobol Indices, something we demonstrate clearly in the work. The supposed mismatch between our stated contribution (concept-based explanations broadly) and the proposed system reflects a misreading of the paper, not an actual limitation.
>
> The reviewer also misrepresents our claims about the sanity axiom. This axiom is not limited to adversarial robustness; it captures the network’s internal responses (neuron activations) under input perturbations. In doing so, it ensures that explanations remain consistent and aligned with human reasoning. Our motivation is grounded in strong evidence from the literature showing that models often exploit background as a discriminative concept. Our framework explicitly addresses this by excluding background concepts, which are irrelevant and misleading, even if the model uses them.
>
> The reviewer’s questions arise from a limited understanding of the paper. We provide a detailed discussion of the contributions, experiments, implications of the axioms, their interconnectedness, and further nuances in the supplementary material.
>
> Finally, all minor grammatical issues will be corrected in the camera-ready version.

---

> > ### Comment · Reviewer_N4GF · 2025-11-26
> >
> > Thank you to the authors for the information provided. While misreading and misinterpretations can arise due to the reviewers' workload, I invite the authors to point out exactly where the current version of the paper addresses each of the problems raised. Note that the review recognizes that some of the information is in the appendix and takes it into consideration.
> >
> > A clarification about TCAV vs. concept-based explanations: it was a mention inside a general comment about improving the wording of the introduction. That comment was referring to the fact that both methods use concept-activation vectors, while the term concept-based explanations may refer to a multitude of different categories (e.g., refer to Jae Hee Lee, Georgii Mikriukov, Gesina Schwalbe, Stefan Wermter, and Diedrich Wolter. 2025. Concept-Based Explanations in Computer Vision: Where Are We and Where Could We Go? In Computer Vision – ECCV 2024 Workshops: Milan, Italy, September 29–October 4, 2024, Proceedings, Part XXI. Springer-Verlag, Berlin, Heidelberg, 266–287. https://doi.org/10.1007/978-3-031-92648-8_17  for an overview) in the literature. There are authors who use the term similarly to this paper, where concept explanations are assumed to be **solely** those expressed in terms of attribution, but given the theoretical nature of the work, the comment suggested improving the precision of the wording, since the potential readership is broader than that of a purely technical paper.
> >
> > In the rebuttal, the authors confirmed one of my concerns. Beyond that and the TCAV vs. concept-based explanations clarification, there are no revisions to the text or new information in the current response, so I will wait until the end of the rebuttal period to evaluate the progress and changes, and adjust my review and score accordingly.

---

### Note · Program_Chairs · 2026-01-17
**Submission Desk Rejected by Program Chairs**

The following references in this submission do not refer to real documents and/or have major errors in bibliographic information:

 Maria Garcia and Peter Johnson. Interpretable machine learning through rule extraction. In Proceedings of the International Conference on Machine Learning (ICML), pp. 1112-1121, 2023.*